# The Percentages of Cognitive Skills Deficits among Chinese Children with Developmental Dyslexia: A Systematic Review and Meta-Analysis

**DOI:** 10.3390/brainsci12050548

**Published:** 2022-04-26

**Authors:** Xin Li, Mingming Hu, Huadong Liang

**Affiliations:** 1School of Information Science and Technology, University of Science and Technology of China, Hefei 230026, China; 2Artificial Intelligence Research Institute, iFLYTEK Co., Ltd., Hefei 230088, China; mmhu6@iflytek.com (M.H.); hdliang@iflytek.com (H.L.)

**Keywords:** developmental dyslexia, cognitive skills deficits, percentage, core deficit

## Abstract

The current study was conducted to examine the percentages of cognitive skills deficits among Chinese children with developmental dyslexia. Via a systematic review, we collated twenty-two available studies on the proportion of cognitive skills deficits, including phonological awareness, rapid automatized naming, morphological awareness, orthographic knowledge, short-term memory and working memory, and visual and motor skills deficits, among Chinese children with developmental dyslexia. The results of a meta-analysis showed that the rapid automatized naming deficits are the core deficit of developmental dyslexia among Chinese children, with a pooled percentage of 44%. This is followed by orthographic knowledge deficits (43%), phonological awareness deficits (41%), morphological awareness deficits (40%), visual and motor skills deficits (33%), and short-term memory and working memory deficits (25%). At the same time, we compared the proportions of different locations, ages, standards and control groups.

## 1. Introduction

Developmental dyslexia, defined as a specific language-based disorder, is not attributable to a disorder of intellectual development, neurological disorder, lack of availability of education, lack of proficiency in the language of academic instruction, or psycho-social adversity. A percentage for developmental dyslexia has been reported as approximately 7% of the general population in western countries [1]. In china, about 5~8% of school-aged children have difficulties in reading Chinese [2,3].

In recent years, researchers have put forward various theories about cognitive deficits in Chinese dyslexia [4,5]. Exploring the cognitive skill deficits of Chinese dyslexia is helpful to understand the cause of children with dyslexia. In developmental dyslexia research, the identification of phonological awareness, rapid automatized naming, orthographic knowledge, morphological awareness, short-term memory and working memory, and visual and motor skills as important factors in learning to read and in the specific reading difficulties of developmental dyslexia reflects the general consensus [6]. Phonological awareness refers to the ability to detect and manipulate the sound structure of words independently of their meaning [7]. The assessment of phonological awareness usually includes rhyme awareness, syllable awareness and phonemic awareness tasks [8]. Rapid automatized naming refers to the ability to name as fast as possible highly familiar stimuli. The tasks include digits, letters, characters, objects/pictures, and colors. Orthographic knowledge refers to the ability to abstract representation of character [9]. The assessment usually includes a character decision task and partial cue-based recognition task. Morphological awareness refers to the awareness of morpheme structures and the ability to manipulate them [10]. The assessment usually includes a morpheme identification task, morphological construction test and homophone production test. Short-term memory and working memory refer to the ability to temporarily retain information. The assessment usually includes phonological memory tasks and digit span tasks. Visual skill refers to a child’s general cognitive abilities. Move skill refers to movements with muscles. The visual attention span task is one of the most common measurement tasks.

A large number of studies in western countries show that phonological processing is the core deficit of developmental dyslexia. However, there are many differences between Chinese and pinyin characters, and researchers have different views on the core deficit of Chinese dyslexia. Ho examined patterns of cognitive deficits in dyslexia and found that 29% of children in the Chinese dyslexia group had phonological deficits, 57% had rapid automatized naming deficits, 42% had orthographic skills deficits and 27% had visual and motor skills deficits [11]. Therefore, rapid automatized naming deficits may be the core deficits in Chinese dyslexia. However, the study by Chung found that 22% of children in the Chinese dyslexia group had phonological deficits, 48% had rapid automatized naming deficits, 78% had orthographic skills deficits, 67% had morphological deficits, and 52% had short-memory deficits [12]. Based on the study of Chung, orthographic skills deficits may be the core deficit. Indeed, some researchers still believe that phonological awareness is the core deficit of Chinese dyslexia. In the study by Liu, 45% had phonological deficits, 41% had rapid automatized naming deficits, 35% had orthographic skills deficits and 14% had morphological deficits [13]. The variability of percentage may also be related to other factors, such as age, control group and location [14].

It can be observed from the above research that Chinese dyslexia has multiple language deficits, but the core deficit is still controversial among researchers. Understanding the core deficits of dyslexia can lead to targeted interventions and treatments. It is also important that clinicians have reliable prevalence estimates to gain an understanding of the proportion of individuals with developmental dyslexia who may meet the criteria for cognitive deficits at a given point in time, in order to appropriately assess and plan tailored treatment to maximize recovery outcomes. The study aim was to conduct a meta-analysis to estimate the percentage of cognitive deficits for developmental dyslexia in Chinese dyslexia. We reported the percentage for the different criteria. Furthermore, the study also wanted to identify which cognitive deficit is the core deficit in Chinese dyslexia.

## 2. Materials and Methods

### 2.1. Search Strategy and Procedure

The articles for this meta-analysis were identified by searching the Web of Science (core collection) and CNKI. The combination of search terms applied included “reading dis* OR reading dif* OR poor read* OR developmental dyslexi*”, “individual difference OR deficit OR subtype” and “Chinese”. The titles, abstracts, keywords and full texts were screened to determine whether the inclusion criteria were met. Databases including Web of Science and CNKI were searched to identify articles from inception to September 18th, 2021. The initial search yielded 2719 articles. The protocol for the systematic review was conceived based on the PRISMA 2020 Statement (Table A1). It was submitted for registration in the PROSPERO international prospective register of systematic reviews (ID: 321448, status: waiting for approval). Two researchers (M.H. and H.L.) independently conducted a literature search. Then, a search of the reference lists of the articles included in the first step was performed to complement our database searches.

### 2.2. Inclusion and Exclusion Criteria

The articles were included if (1) the type of study was experimental, including a group of native Chinese-speaking people with DD(reading disability, reading disorder, reading difficulties, poor reading, developmental dyslexia); (2) the percentage of cognitive deficits for Chinese developmental dyslexia were reported or can be calculated; (3) published in English or Chinese; (4) the studies are not duplicated in the existing literature; (5) different literatures came from the same sample, and the results with the most comprehensive reports and up-to-date data were selected. Articles were excluded if they were conference papers, review papers or qualitative studies. In addition, unpublished papers were not included, due to the difficulty of obtaining the full text and detailed information.

### 2.3. Recorded Variables and Coding

#### 2.3.1. Coding Procedure

The variables were discussed until a consensus was reached among all the authors. Then, two raters used the recorded variables to conduct the coding of all the articles. Across the total variable matrix, the mean inter-rater agreement coefficient (M.H. and H.L.) was 0.96. Any disagreements between raters were resolved by discussion with the third person (X.L.).

#### 2.3.2. Variables

For each study, the following variables were recorded: (1) the sample characteristics; (2) the definition criteria of cognitive deficits; (3) the type of cognitive deficits and percentages of different cognitive deficits. It is important to note that different cognitive skills may be measured using different tasks in different studies. In order to minimize the impact of the tasks, when a cognitive skill involved multiple tasks for evaluation, the average percentage in the various tasks was selected. Table 1 shows a detailed explanation of the variables.

### 2.4. Statistical Analysis

In the study, we used Stata Statistical 15.0 software, including summary estimation, forest mapping and publication bias assessment. If the heterogeneity was low (*p* > 0.1, I^2^ ≤ 50%), the fixed effects model was selected for analysis. Otherwise, the random effects model was selected. A subgroup analysis was also performed to explore the possible sources of heterogeneity among studies. Publication bias was established based on the funnel plot and Egger test. For the meta-analysis results, *p* < 0.05 was considered as statistically significant.

## 3. Results

Figure 1 shows the main process of literature search and study selection. The search yielded 2719 records. A total of 22 articles including twenty-six studies met the study inclusion criteria and were included in this meta-analysis (Figure A1). Among these, one reported the results of sample tracking twice, so we obtained two results. One study reported results from the same sample compared to two different control groups, so we recoded the two results. In addition, one study with three different age groups recorded three results.

### 3.1. Study Characteristics

Twenty-two articles, including twenty-six studies, met the criteria for inclusion in this meta-analysis and are listed in Table 2. The combined sample size of dyslexia in all the studies was 1284, while the individual study sample of dyslexia ranged from 15 to 223 participants. The studies were all conducted in China, including Hong Kong, from 2002 to 2021, and all the participants speak Chinese as a first language. Eighteen of the studies reported the percentages of phonological awareness deficits, sixteen reported the percentages of rapid automatized naming deficits, eleven reported the percentages of orthographic knowledge deficits, ten reported the percentages of morphological awareness deficits, ten reported the percentages of short-memory deficits and nine reported the percentages of visual and motor skills deficits.

Publication bias was established based on the funnel plot (Figure A2) and Egger test (t = 0.81, *p* = 0.432, for phonological awareness; t = 0.73, *p* = 0.478, for rapid automatized naming; t = 0.30, *p* = 0.769 for orthographic knowledge; t = −0.7, *p* = 0.502, for morphological awareness; t = 1.11, *p* = 0.291, for short-term memory and working memory; t = 0.65, *p* = 0.531, for visual and motor skills).

### 3.2. Pooled Percentage 

We used I^2^ to test the heterogeneity between studies. If the heterogeneity was low (*p* > 0.1, I^2^ ≤ 50%), the fixed-effect model was selected to estimate pooled percentage, otherwise, the random effect model was selected. 

#### 3.2.1. Phonological Awareness

Table 3 showed the results of overall and subgroup meta-analysis about phonological awareness deficits. The percentages of phonological awareness deficits range from 9% to 76%, with a pooled percentage of 41% (95% CI: 31–52%). 

For age, we divided the sample into two groups, with a cut-off age of 11. For the age group of children younger than 11 years old, 11 studies reported a pooled percentage of 41% (95% CI: 27–55%). The remaining six studies reported a pooled percentage of 38% (95% CI: 19–56%). For the type of areas, 12 studies reported ae pooled percentage of 49% (95% CI: 42–56%), with the sample from Mainland China. In addition, six studies reported a pooled percentage of 26% (95% CI: 5–47%), with the sample from Hong Kong, China. For the type of control group, fourteen studies used age-matched typically developing children as controls to confirm whether children with DD have phonological awareness deficits and reported a pooled percentage of 37% (95% CI: 26–47%). Two studies used reading-level-matched typically developing children as controls to confirm whether children with DD have phonological awareness deficits and reported a pooled percentage of 49% (95% CI: 38–59%). In addition, two studies used a cluster analysis and reported a pooled percentage of 64% (95% CI: 41–87%). For the criterion of deficits, ten studies used the cut-off criteria of 1.5 standard deviations below the mean on phonological awareness deficits screening and reported a pooled percentage of 36% (95% CI: 24–48%). Three studies used the cut-off criteria of 1 standard deviations below the mean and reported a pooled percentage of 31% (95% CI: 5–57%). Two studies used the cluster method and reported a pooled percentage of 64% (95% CI: 41–87%). A study used the criteria of 1.65 standard deviations below the mean and reported a pooled percentage of 47% (95% CI: 21–72%), a study used the criteria of 2 standard deviations below the mean and reported a pooled percentage of 40% (95% CI: 21–59%), and a study used the cut-off criteria of the mean and reported a pooled percentage of 73% (95% CI: 51–96%).

#### 3.2.2. Rapid Automatized Naming

Table 4 showed the results of overall and subgroup meta-analysis about rapid automatized naming deficits. The percentages of rapid automatized naming deficits range from 17% to 66%, with a pooled percentage of 44% (95% CI: 37–51%).

In the age group of children younger than 11 years old, nine studies reported a pooled percentage of 46% (95% CI: 36–56%). The remaining seven studies reported a pooled percentage of 42% (95% CI: 32–53%). For the type of areas, nine studies reported a pooled percentage of 36% (95% CI: 29–43%), with the sample from Mainland China. In addition, seven studies reported a pooled percentage of 56% (95% CI: 51–61%), with the sample from Hong Kong, China. For the type of control group, 13 studies used age-matched typically developing children as controls to confirm whether children with DD have rapid automatized naming deficits and reported a pooled percentage of 48% (95% CI: 41–54%). Two studies used reading-level-matched typically developing children as controls to confirm whether children with DD have rapid automatized naming deficits and reported a pooled percentage of 28% (95% CI: 5–52%). In addition, one study used a cluster analysis and reported a pooled percentage of 37% (95% CI: 31–44%). For the criterion of deficits, eleven studies used the cut-off criteria of 1.5 standard deviations below the mean on rapid automatized naming deficits screening and reported a pooled percentage of 44% (95% CI: 34–54%). Four studies used the cut-off criteria of 1 standard deviations below the mean and reported a pooled percentage of48% (95% CI: 35–61%). A study used the cluster method and reported a pooled percentage of 37% (95% CI: 31–44%).

#### 3.2.3. Orthographic Knowledge

Table 5 showed the results of overall and subgroup meta-analysis about orthographic knowledge deficits. The percentages of orthographic knowledge deficits range from 27% to 64%, with a pooled percentage of 43% (95% CI: 36–50%). 

In the age group of children younger than 11 years old, eight studies reported a pooled percentage of 40% (95% CI: 32–49%). The remaining three studies reported a pooled percentage of 51% (95% CI: 41–62%). For the type of areas, four studies reported a pooled percentage of 46% (95% CI: 29–64%), with the sample from Mainland China. In addition, seven studies reported a pooled percentage of 41% (95% CI: 34–49%), with the sample from Hong Kong, China. For the type of control group, eight studies used age-matched typically developing children as controls to confirm whether children with DD have orthographic knowledge deficits and reported a pooled percentage of 43 % (95% CI: 35–51%). Two studies used reading-level-matched typically developing children as controls to confirm whether children with DD have orthographic knowledge deficits and reported a pooled percentage of 31 % (95% CI: 22–41%). Furthermore, one study used a cluster analysis and reported a pooled percentage of 57 % (95% CI: 47–68%). For the criterion of deficits, seven studies used the cut-off criteria of 1.5 standard deviations below the mean on orthographic knowledge deficits screening and reported a pooled percentage of 43% (95% CI: 35–52%). Three studies used the cut-off criteria of 1 standard deviations below the mean and reported a pooled percentage of 33 % (95% CI: 24–42%). A study used the cluster method and reported a pooled percentage of 57 % (95% CI: 47–68%).

#### 3.2.4. Morphological Awareness

Table 6 showed the results of overall and subgroup meta-analysis about morphological awareness deficits. The percentages of morphological awareness deficits range from 12% to 76%, with a pooled percentage of 40% (95% CI: 24–55%). The study with a percentage of 100% was excluded from the actual meta-analysis.

In the age group of children younger than 11 years old, four studies reported a pooled percentage of 24% (95% CI: 0–48%). The remaining six studies reported a pooled percentage of 51% (95% CI: 35–67%). For the type of areas, the seven studies reported a pooled percentage of 37% (95% CI: 18–57%), with the sample from Mainland China. In addition, three studies reported a pooled percentage of 47% (95% CI: 23–71%), with the sample from Hong Kong, China. For the type of control group, seven studies used age-matched typically developing children as controls to confirm whether children with DD have morphological awareness deficits and reported a pooled percentage of 46% (95% CI: 28–64%). Two studies used reading-level-matched typically developing children as controls to confirm whether children with DD have morphological awareness deficits and reported a pooled percentage of 13% (95% CI: 6–20%). Furthermore, one study used a cluster analysis and reported a pooled percentage of 53% (95% CI: 47–60%). For the criterion of deficits, five studies used the cut-off criteria of 1.5 standard deviations below the mean on morphological awareness deficits screening and reported a pooled percentage of 44% (95% CI: 16–73%). Three studies used the cut-off criteria of 1 standard deviations below the mean and reported a pooled percentage of 35% (95% CI: 11–60%). A study used the cut-off criteria of 2 standard deviations below the mean and reported a pooled percentage of 16% (95% CI: 2–30%). A study used the cluster method and reported a pooled percentage of 53% (95% CI: 47–60%).

#### 3.2.5. Short-Term Memory and Working Memory

Table 7 showed the results of overall and subgroup meta-analysis about short-term memory and working memory deficits. The percentages of short-term memory and working memory deficits range from 10% to 52%, with a pooled percentage of 25% (95% CI: 18–31%).

In the age group of children younger than 11 years old, six studies reported a pooled percentage of 23% (95% CI: 13–33%). The remaining six studies reported a pooled percentage of 26% (95% CI: 17–36%). For the type of areas, five studies reported a pooled percentage of 28% (95% CI: 13–43%), with the sample from Mainland China. In addition, seven studies reported a pooled percentage of 23% (95% CI: 16–30%), with the sample from Hong Kong, China. For the type of control group, all twelve studies used age-matched typically developing children as controls to confirm whether children with DD have short-term memory and working memory deficits and reported a pooled percentage of 25% (95% CI: 18–31%). For the criterion of deficits, eight studies used the cut-off criteria of 1.5 standard deviations below the mean on short-term memory and working memory deficits screening and reported a pooled percentage of 26% (95% CI: 16–35%). Three studies used the cut-off criteria of 1 standard deviations below the mean and reported a pooled percentage of 25% (95% CI: 17–33%). A study used the cut-off criteria of 2 standard deviations below the mean and reported a pooled percentage of 16% (95% CI: 2–30%).

#### 3.2.6. Visual and Motor Skills

Table 8 showed the results of overall and subgroup meta-analysis about visual and motor skills deficits. The percentages of visual and motor skills deficits range from 5% to 65%, with a pooled percentage of 33% (95% CI: 20–46%).

In the age group of children younger than 11 years old, eight studies reported a pooled percentage of 31% (95% CI: 16–45%). The remaining one study reported a pooled percentage of 39% (95% CI: 16–61%). For the type of areas, five studies reported a pooled percentage of 35% (95% CI: 12–58%), with the sample from Mainland China. In addition, four studies reported a pooled percentage of 29% (95% CI: 23–35%), with the sample from Hong Kong, China. For the type of control group, eight studies used age-matched typically developing children as controls to confirm whether children with DD have deficits on visual and motor skills and reported a pooled percentage of 29% (95% CI: 18–40%). One study used reading-level-matched typically developing children as controls to confirm whether children with DD have visual and motor skills deficits and reported a pooled percentage of 24% (95% CI: 7–41%). In addition, one study used a cluster analysis and reported a pooled percentage of 65% (95% CI: 54–76%). For the criterion of deficits, five studies used the cut-off criteria of 1.5 standard deviations below the mean on visual and motor skills deficits screening and reported a pooled percentage of 31% (95% CI: 25–36%). Three studies used the cut-off criteria of 1.65 standard deviations below the mean and reported a pooled percentage of 15% (95% CI: −1–31%). A study used the cut-off criteria of the mean and reported a pooled percentage of 53 % (95% CI: 28–79%), and a study used the cluster method and reported a pooled percentage of 65 % (95% CI: 54–76%).

## 4. Discussion

After conducting a meta-analysis of all the available studies that adhered to our inclusion criteria (22 articles), we calculated the pooled percentages under different categories.

### 4.1. Pooled Percentage

We found that the rapid automatized naming deficits are the core deficit of Chinese developmental dyslexia, with a pooled percentage of 44% through meta-analysis. This is followed by orthographic knowledge deficits (43%), phonological awareness deficits (41%), morphological awareness deficits (40%), visual and motor skills deficit (33%), and short-term memory and working memory deficits (25%).

It can be observed from the results that the incidence of rapid automatized naming deficits and orthographic knowledge deficits is relatively high in Chinese dyslexia. In a recent meta-analysis on the deficit profiles of Chinese children with reading difficulties, Peng et al. found that rapid automatized naming deficits and orthographic knowledge deficits may have a greater impact on developmental dyslexia than on any other skill deficits [6]. This is similar to our results. Many studies have shown that rapid automatized naming has a strong predictive effect on developmental dyslexia and can effectively identify developmental dyslexia [11,36]. Rote learning is usually the main method of learning Chinese character, and it is a way of learning that may have led to rapid automatized naming skills as the basis of Chinse character acquisition [37]. Based on Wolf’s idea, rapid automatized naming tasks are complex and involve cognitive perceptual and linguistic processes [38]. Therefore, children with rapid automatized naming deficits may also have deficits in orthographic knowledge deficits. In addition, according to previous studies, the orthographic knowledge of Chinese reading involves determining the pronunciation of Chinese characters according to the phonetic element radicals, obtaining the semantics based on radicals and grasping the overall structure of Chinese characters. For children with Chinese developmental dyslexia, it takes more time and effort to acquire these complex rules. So, rapid automatized naming and orthographic knowledge skills may be the most important Chinese reading skills [18]. Unlike the language of the West, phonological awareness deficits do not show a higher incidence in Chinese dyslexia. Chinese characters are semiotic characters, and their form and meaning are closely related, so the causes of Chinese dyslexia may be more complicated [17].

### 4.2. Type of Control Group

Compared to the age-matched typically developing children, children with dyslexia have a higher percentage of rapid automatized naming deficit (48%). This is followed by morphological awareness deficits (46%), orthographic knowledge deficits (43%), phonological awareness deficits (37%), visual and motor skills deficits (29%), and short-term memory and working memory deficits (25%). However, compared to the reading-level-matched typically developing children, children with dyslexia have a higher percentage of phonological awareness deficits (49%). This is followed by orthographic knowledge deficits (31%), rapid automatized naming deficits (28%), visual and motor skills deficits (24%), and morphological awareness deficit (13%). According to the existing results, the percentage of rapid automatized naming deficits and orthographic knowledge deficits was relatively high, when the control group was age-matched typically developing children or reading-level-matched typically developing children. In addition, the percentage of visual and motor skills deficits and short-term memory and working memory deficits was relatively low. Since reading is a language activity, the deficits of dyslexia children were mainly related to reading language skills. So, researchers paid more attention to the linguistic cognitive deficits of dyslexia, such as phonological awareness deficits, orthographic knowledge deficits and rapid automatized naming deficits. However, in recent years, visual deficits have also been proposed as the core deficit of dyslexia. Bosse found in two studies of people in France and Britain that dyslexia did not seem to be due to phonological deficits and the visual attention deficit is likely to be the underlying cause of dyslexia [39]. Franceschini et al. found that visual spatial attention in preschool children could predict future reading acquisition [40]. Although the results of the study cannot prove the importance of basic cognitive skills, the explanation of the causes of dyslexia should be found from more perspectives to find deeper reasons.

### 4.3. Age, Location and Standard

Studies have shown that age may influence the deficit profile of children with dyslexia [41]. According to the results of our study, the percentage of cognitive skill deficits in different age groups is relatively close, except for the relatively large difference in morphological awareness deficits (24% vs. 51%). We found that there was an imbalanced development of morphological awareness. This reminds us to pay more attention to the development of morphological awareness in the lower grades and intervene in time to avoid morphological awareness deficits in the higher grades. Although some studies also found that age may influence the deficit profiles of rapid automatized naming [6], we did not find a significant difference between the two age groups, in terms of proportion of occurrence. It is possible that the sample size we have at present is relatively narrow in age range and the span is not large enough. Therefore, it is necessary to further study the interaction between age and cognitive skills.

Location may also be the reason for the difference in the incidence of cognitive deficits among dyslexic groups, as there are still many differences in spoken language, writing scripts and early reading instructions between Mainland China and Hong Kong [6]. The percentages of phonological awareness deficits (49% vs. 26%), rapid automatized naming deficits (36% vs. 56%), morphological awareness deficits (37% vs. 47%) between Mainland and Hong Kong have a relatively large difference. Although some studies suggested that the education environment is similar between Mainland China and Hong Kong [6], the children from Hong Kong may be more familiar with English than the children from Mainland China. This may have a certain effect on the skill deficits of dyslexia.

The differences in the definition of a skill deficit may also lead to differences in incidence. Although most studies used standard deviation segmentation, some used 1 standard deviation lower than the control group, while others used 1.5 standard deviation or 2 standard deviation lower than the control group. However, this study did not find a trend of decreasing incidence with the stricter standards, which may be due to the fact that most of the existing studies were based on the cut-off score of 1.5 standard deviations, while the sample size of other standards was limited.

### 4.4. Limitations

Our findings are only based on the combined results of 22 articles, which is a small number of studies for a meta-analysis. This may be due to our poor search coverage and stringent screening criteria, which also reduce the reliability of the findings. In particular in the subgroup analysis, many groups involved only one study, which brings great challenges to the reliability of our research results. In addition, we paid more attention to language cognitive skills and general cognitive skills that affect developmental dyslexia, while higher-order cognitive skills, such as creativity, were not involved. However, some studies have found that dyslexia may be related to higher levels of creativity [42,43]. Therefore, higher-order cognitive skills that affect developmental dyslexia may also need further exploration.

## 5. Conclusions

The present study is the first meta-analysis to systematically investigate the core deficit among Chinese children with developmental dyslexia. Based on the above analysis, we found that the rapid automatized naming deficits are the core deficit of Chinese developmental dyslexia. In addition, the pooled percentages of orthographic knowledge deficits, phonological awareness deficits, and morphological awareness deficits among Chinese children with dyslexia are also relatively higher. The pooled percentages of short-term memory and visual and motor skills deficit are relatively lower. These findings could have important implications for the screening of developmental dyslexia. The accuracy of diagnosis could be improved through the measurement of cognitive skills of developmental dyslexia. Moreover, in the daily teaching of Chinese, we should emphasize rapid automatized naming, orthographic knowledge and phonological awareness and strengthen skills training to reduce the incidence of developmental dyslexia. Certainly, the findings support the multiple-deficit hypothesis in Chinese developmental dyslexia.

## Figures and Tables

**Figure 1 brainsci-12-00548-f001:**
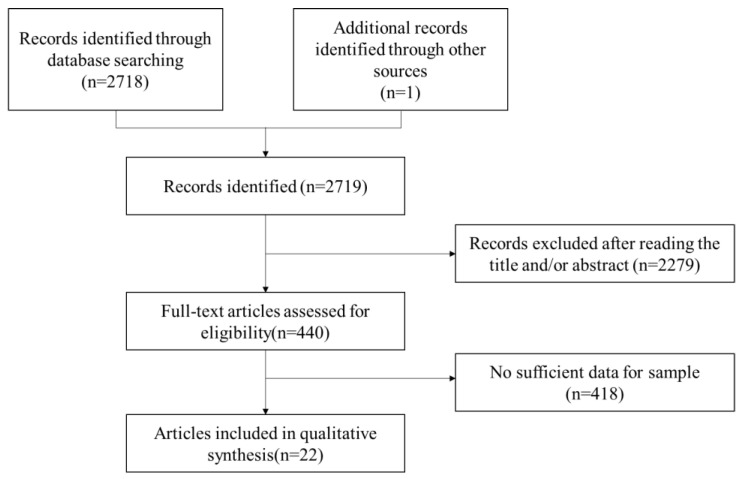
Flow diagram of the literature search and study selection.

**Table 1 brainsci-12-00548-t001:** The detailed explanation of the variables.

Variables	Contents	Specific Description
The Sample Characteristics	Sample size	The number of people with Chinese developmental dyslexia was coded.
	Age	The mean age of the sample was coded.
The Definition Criteria of Cognitive Deficits	Type of Control Group	The researchers used age-matched typically developing or reading-level-matched typically developing children as controls to further confirm whether children with DD have certain deficits [15,16]. The type of control was coded.
	Criterion of Cognitive deficits	The children were identified as having a cognitive deficit if their performance was below the cut-off criteria of the control group (e.g., 1.5SD below the mean of a participant’s respective age group) on the cognitive deficits screening measures.
The Type of Cognitive Deficits and Percentage of Different Cognitive Deficits	Type of Cognitive Deficits	The cognitive deficits included phonological awareness, rapid automatized naming, orthographic knowledge, morphological awareness, short-term memory and working memory, and visual and motor skills. If there was a cognitive deficit in the paper that did not fall into any of the above categories, it went into the other category.
	The Percentage of Cognitive Deficits	The percentage or the sample size of different cognitive deficits was coded.

**Table 2 brainsci-12-00548-t002:** Characteristics of the included studies, examining the percentage of cognitive skill deficits for the dyslexia group.

Research	Location	Dyslexia (n)	Age (Years)	Control Group	Standard	Phonological Awareness Deficits	Rapid Automatized Naming Deficits	Orthographic Knowledge Deficits	Morphological Awareness Deficits	Short-Term Memory and Working Memory Deficits	Visual and Motor Skills Deficits
Liu, Liu and Zhang, 2006 [13]	Mainland	29	10.43	RC	1 SD	0.448	0.414	0.345	0.138		
Xiong and Yan, 2014 [17]	Mainland	57	10.58	RC	1.5 SD	0.509	0.175	0.298	0.123		
Wang, Li and Deng, 2014 [18]	Mainland	33	11.54	CA	1.5 SD	0.697	0.348	0.636		0.364	
Wu, Shu and Wang, 2004 [19]	Mainland	15	11.6	CA	1.65 SD	0.467			1		
Peng et al., 2007 [20]	Mainland	25	3.5	CA	2 SD	0.4			0.16	0.16	
Li and Shu, 2009 [21]	Mainland	41	11.7	CA	1 SD	0.415	0.341		0.512	0.354	
Meng, Zhou, Zeng, Kong and Zhuang, 2002 [22]	Mainland	15	10–11.5	CA	Mean	0.733					0.533
Wu, Shu and Liu, 2005 [23]	Mainland	91	11.83–12.17	CA	1.5 SD	0.429	0.407		0.758		
Chen, Yang and Tang, 2002 [24]	Mainland	77	10	Cluster	Cluster						0.649
Ho et al., 2004 [11]	Hong Kong	147	8.275	CA	1.5 SD	0.252	0.571	0.42		0.321	0.272
Ho, Chan, Tsang and Lee, 2002 [25]	Hong Kong	30	8.67	CA	1.5 SD	0.167	0.5	0.389		0.1335	0.367
Chen, Zheng and Ho, 2019 [26]	Hong Kong	25	10.45	CA	1.5 SD						0.4
	Hong Kong	25	10.45	RC	1.5 SD						0.24
Song, Zhang, Shu, Su and McBride, 2020 [27]	Mainland	223	10.84	Cluster	Cluster	0.525	0.372		0.534		
Huo, Wu, Mo, Wang and Maurer, 2021 [28]	Hong Kong	84	8.39	Cluster	Cluster	0.762		0.571			
Li, Shu, McBride-Chang, Liu and Xue, 2009 [29]	Mainland	41	11.73	CA	1.5 SD	0.22	0.268		0.366	0.0976	
Chung et al., 2010 [30]	Hong Kong	27	13.65	CA	1.5 SD	0.074	0.352	0.407	0.296	0.259	
Chung, Lo, Ho, Xiao and Chan, 2014 [31]	Hong Kong	52	13.42	CA	1.5 SD		0.61		0.67	0.33	
Wang, Georgiou, Das and Li, 2012 [32]	Mainland	27	9.98	CA	1.5 SD	0.5185	0.4445	0.5926		0.5185	
Chung, Lo and McBride, 2018 [12]	Hong Kong	50	9.04	CA	1 SD	0.09	0.52	0.267		0.207	
	Hong Kong	25	13.31	CA	1 SD		0.66	0.46	0.42	0.26	
Chan, Hung, Liu and Lee, 2008 [33]	Hong Kong	43	8.17	CA	1.5 SD	0.233	0.628	0.372		0.116	
Zhao, Liu, Liu and Huang, 2018 [34]	Mainland	20	8.88	CA	1.65 SD						0.1
	Mainland	19	10.19	CA	1.65 SD						0.0526
	Mainland	18	11.68	CA	1.65 SD						0.3889
Cheng, Yao, Wang and Zhao 2021 [35]	Mainland	45	10.11	CA	1.5 SD	0.6	0.533				0.4

Abbreviations: CA = age-matched typically developing children as controls; RC = reading-level-matched typically developing children as controls; SD = standard deviation.

**Table 3 brainsci-12-00548-t003:** The percentage of phonological awareness deficits for dyslexia group.

Index	Number of Studies	Heterogeneity Test		Model	Results	
		*p*	*I^2^*		Pooled Percentage	95% CI
**Total**	18	<0.001	93.20%	random	0.41	(0.31, 0.52)
**Age**						
Younger than 11 years old	11	<0.001	94.50%	random	0.41	(0.27, 0.55)
Older than 11 years old	6	<0.001	91.00%	random	0.38	(0.19, 0.56)
**Location**						
Mainland	12	<0.001	69.10%	random	0.49	(0.42, 0.56)
Hong Kong	6	<0.001	96.70%	random	0.26	(0.05, 0.47)
**Control group**						
CA	14	<0.001	89.70%	random	0.37	(0.26, 0.47)
RC	2	0.591	0.00%	fixed	0.49	(0.38, 0.59)
**Standard**						
mean	1	—	—	fixed	0.73	(0.51, 0.96)
1SD	3	<0.001	91.10%	random	0.31	(0.05, 0.57)
1.5SD	10	<0.001	90.00%	random	0.36	(0.24, 0.48)
1.65SD	1	—	—	fixed	0.47	(0.21, 0.72)
2SD	1	—	—	fixed	0.4	(0.21, 0.59)
Cluster	2	<0.001	94.20%	random	0.64	(0.41, 0.87)

Abbreviations: CA= age-matched typically developing children as controls; RC= reading-level-matched typically developing children as controls; SD = standard deviation.

**Table 4 brainsci-12-00548-t004:** The percentage of rapid automatized naming deficits for the dyslexia group.

Index	Number of Studies	Heterogeneity Test		Model	Results	
		*p*	*I^2^*		Pooled Percentage	95% CI
**Total**	16	<0.001	79.70%	random	0.44	(0.37, 0.51)
**Age**						
Younger than 11 years old	9	<0.001	84.70%	random	0.46	(0.36, 0.56)
Older than 11 years old	7	<0.001	80.70%	random	0.42	(0.32, 0.53)
**Location**						
Mainland	9	0.004	64.80%	random	0.36	(0.29, 0.43)
Hong Kong	7	0.207	29.10%	fixed	0.56	(0.51, 0.61)
**Control group**						
CA	13	<0.001	67.20%	random	0.48	(0.41, 0.54)
RC	2	0.022	80.90%	random	0.28	(0.05, 0.52)
**Standard**						
1SD	4	0.047	62.20%	random	0.48	(0.35, 0.61)
1.5SD	11	<0.001	83.80%	random	0.44	(0.34, 0.54)
Cluster	1	—	—	fixed	0.37	(0.31, 0.44)

Abbreviations: CA = age-matched typically developing children as controls; RC = reading-level-matched typically developing children as controls; SD = standard deviation.

**Table 5 brainsci-12-00548-t005:** The percentage of orthographic knowledge deficits for the dyslexia group.

Index	Number of Studies	Heterogeneity Test		Model	Results	
		*p*	*I^2^*		Pooled Percentage	95% CI
**Total**	11	0.001	65.80%	random	0.43	(0.36, 0.50)
**Age**						
Younger than 11 years old	8	0.002	68.40%	random	0.4	(0.32, 0.49)
Older than 11 years old	3	0.158	45.90%	fixed	0.51	(0.41, 0.62)
**Location**						
Mainland	4	0.002	79.60%	random	0.46	(0.29, 0.64)
Hong Kong	7	0.024	58.70%	random	0.41	(0.34, 0.49)
**Control group**						
CA	8	0.02	58.10%	random	0.43	(0.35, 0.51)
RC	2	0.661	0.00%	fixed	0.31	(0.22, 0.41)
**Standard**						
1SD	3	0.254	27.00%	fixed	0.33	(0.24, 0.42)
1.5SD	7	0.024	58.90%	random	0.43	(0.35, 0.52)
Cluster	1	—	—	fixed	0.57	(0.47, 0.68)

Abbreviations: CA= age-matched typically developing children as controls; RC= reading-level-matched typically developing children as controls; SD = standard deviation.

**Table 6 brainsci-12-00548-t006:** The percentage of morphological awareness deficits for dyslexia group.

Index	Number of Studies	Heterogeneity Test		Model	Results	
		*p*	*I^2^*		Pooled Percentage	95% CI
**Total**	10	<0.001	94.50%	random	0.4	(0.24, 0.55)
**Age**						
Younger than 11 years old	4	<0.001	96.00%	random	0.24	(0.00, 0.48)
Older than 11 years old	6	<0.001	87.30%	random	0.51	(0.35, 0.67)
**Location**						
Mainland	7	<0.001	96.00%	random	0.37	(0.18, 0.57)
Hong Kong	3	0.002	84.40%	random	0.47	(0.23, 0.71)
**Control group**						
CA	7	<0.001	91.30%	random	0.46	(0.28, 0.64)
RC	2	0.846	0%	fixed	0.13	(0.06, 0.2)
**Standard**						
1SD	3	0.001	86.80%	random	0.35	(0.11, 0.60)
1.5SD	5	<0.001	97.50%	random	0.44	(0.16, 0.73)
2SD	1	—	—	fixed	0.16	(0.02, 0.30)
Cluster	1	—	—	fixed	0.53	(0.47, 0.60)

Abbreviations: CA = age-matched typically developing children as controls; RC = reading-level-matched typically developing children as controls; SD = standard deviation.

**Table 7 brainsci-12-00548-t007:** The percentage of short-term memory and working memory deficits for dyslexia group.

Index	Number of Studies	Heterogeneity Test		Model	Results	
		*p*	*I^2^*		Pooled Percentage	95% CI
**Total**	12	<0.001	71.20%	random	0.25	(0.18, 0.31)
**Age**						
Younger than 11 years old	6	<0.001	78.70%	random	0.23	(0.13, 0.33)
Older than 11 years old	6	0.012	65.80%	random	0.26	(0.17, 0.36)
**Location**						
Mainland	5	<0.001	81.80%	random	0.28	(0.13, 0.43)
Hong Kong	7	0.013	62.80%	random	0.23	(0.16, 0.30)
**Control group**						
CA	12	<0.001	71.20%	random	0.25	(0.18, 0.31)
RC	0					
**Standard**						
1SD	3	0.49	0.00%	fixed	0.25	(0.17, 0.33)
1.5SD	8	<0.001	80.30%	random	0.26	(0.16, 0.35)
2SD	1	—	—	fixed	0.16	(0.02, 0.30)

Abbreviations: CA= age-matched typically developing children as controls; RC= reading-level-matched typically developing children as controls; SD = standard deviation.

**Table 8 brainsci-12-00548-t008:** The percentage of visual and motor skills deficits for dyslexia group.

Index	Number of Studies	Heterogeneity Test		Model	Results	
		*p*	*I^2^*		Pooled Percentage	95% CI
**Total**	10	<0.001	89.00%	random	0.33	(0.20, 0.46)
**Age**						
Younger than 11 years old	8	<0.001	91.00%	random	0.31	(0.16, 0.45)
Older than 11 years old	1	_	_	fixed	0.39	(0.16, 0.61)
**Location**						
Mainland	5	<0.001	93.70%	random	0.35	(0.12, 0.58)
Hong Kong	4	0.451	0.00%	fixed	0.29	(0.23, 0.35)
**Control group**						
CA	8	<0.001	79.90%	random	0.29	(0.18, 0.40)
RC	1	_	_	fixed	0.24	(0.07, 0.41)
**Standard**						
mean	1	—	—	fixed	0.53	(0.28, 0.79)
1.5SD	5	0.335	12.40%	fixed	0.31	(0.25, 0.36)
1.65SD	3	0.028	72.10%	random	0.15	(−0.01, 0.31)
Cluster	1	—	—	fixed	0.65	(0.54, 0.76)

Abbreviations: CA = age-matched typically developing children as controls; RC = reading-level-matched typically developing children as controls; SD = standard deviation.

## Data Availability

All data related to the research are presented in the article.

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
