# Peer review of "The Percentages of Cognitive Skills Deficits among Chinese Children with Developmental Dyslexia: A Systematic Review and Meta-Analysis"

_brainsci, 2022, doi:10.3390/brainsci12050548_

Round 1
Reviewer 1 Report
I found this submission quite difficult to read. This was partly a function of the poor writing quality, and I have recommended to Editor that any revision will need to be scrutinised by a highly competent English speaker.
There were two other major problems with the manuscript.
a) There is practically no information about the L1 of the participants. They all speak "Chinese", but there are significant differences between mainland varieties and Hong Kong varieties which probably impact on the pareticipants' dyslexia. My guess would be that the Hong Kong dyslexics had at least some familiarity with English, which would also be expected to have an effect that might not be apparent in the mainland dyslexics.
b) More importantly, we have no information about the tests used to assess the various aspects of the dyslexics' performance. My guess would be that readers of Brain Sciences will be unfamiliar with these testing tools, so some explanation needs to be added to the text to make it more user friendly. Furthermore, it seems unlikely that the 21 papers in the review all use identical methods, and this makes it very difficult to make sense of the comparisons. Basically, it is not very illuminating to say that Group X scores 20% on a cognitive skills test, while Group Y scores 45% on a phonological awareness test if the two tests are using different measures of the cognitive skill. (It might be necessary to be more explicit about why phonological awareness is a core skill in dyslexia studies. Readers of Brain Sciences probably won't know this.)
A couple of other points:
c) You need to add Systematic Review to the title of the paper. This type of research is becoming increasingly important, but the search engines will miss your paper if you don't flag up the methodology more clearly.
d) Your discussion section does not pay much attention to what your findings actually mean. For example, you state that a phonological deficit of 45% emerges from the analysis, but what are the implications of this? why does it matter that the phonological awareness feature is only second in the weighted list, and only second by one percentage point. (Is one percentage point on a phonological awareness test "the same" as one percentage point on a test of rapid automatised naming?)
e) A search of Google Scholar shows that there are very many systematic reviews of research in dyslexia. It would have been interesting to know how your results compared with a few of these other studies.
Reviewer 2 Report
This is an interesting systematic review examining the percentages of cognitive skills deficits among Chinese children with developmental dyslexia. I believe that the paper is promising and will have important contribution to the field. I have several concerns that require the attention of the authors:
- It will be important for the authors to elaborate on how additional records identified through other sources were identified. Did the authors use any other search strategy to ensure that they didn't miss any records?
- There should be an elaboration on how the screening was conducted. Was it conducted by 2 screener?
- The inter-rater reliability of the coding procedure should be reported.
- I was unable to locate the publication bias analysis and Egger test in the result section. This is an important test to report.
- It is unclear why the authors did not consider unpublished sources such as conference paper. This should be clarified further.
- One important limitation of the current study is on the lack of consideration of creativity as an cognitive outcome. The positive relationship between dyslexia and creativity has been widely discussed and examined. It is unclear why it was excluded in the current systematic review. This should be discussed in the discussion or limitation section (see Majeed et al., 2021; Cockcroft & Hartgill, 2004). Relevant papers to discuss: Majeed, N. M et al. (2021). Developmental dyslexia and creativity: A meta‐analysis. Dyslexia, 27(2), 187-203. Cockcroft, K., & Hartgill, M. (2004). Focusing on the abilities in learning disabilities: Dyslexia and creativity. Education as Change, 8(1), 61-79.
- Minor comments: There are several typos and grammatical errors that should be rectified. For example, even in the abstract: The current study was conducted to examine the percentages of cognitive skills deficits of among Chinese children with developmental dyslexia. The authors should carefully recheck their manuscript again.
Reviewer 3 Report
I think that this is a very well conducted review (though admittedly, the sample size is small). I think that the manuscript is publishable if the authors can address the followng issues:
- The authors need to strengthen the argument for the review. Why is it necessary for readers to know the percentages of cognitive skills deficit among Chines children with dyslexia. I understand that Chinese childre may have differen profiles of deficits because the so-called deficts are deficts with reference to the kind of language-related performances children are expected to undertake in differen contexts. I would not regard the so-called non-Chinese studies are homogenous (as they involved some many different languages) but it is possible that Chinese children have different kinds of deficits in comparison with non-Chinese children. But why are these differences important for readers of this journal? I think that the authors need to present a convincing argument to motivate the review.
- I am not sure if we can call the review a meta analysis. Metanalysis is for me to identify whether we have suffcient evidence for the effectiveness of some intervention measures or something similar. In the review, the authors mainly identified different types of defict and worked out their percentages. But I also do not now what can be it called if we do not call it a metaanalysis.
- I think that the authors need to highlight how readers can act upon the results of the analysis.What important messages can we take home with in relation to our research and practice?
Round 2
Reviewer 2 Report
The authors have sufficiently addressed my comments.